# Impact Damage Resistance and Post-Impact Tolerance of Optimum Banana-Pseudo-Stem-Fiber-Reinforced Epoxy Sandwich Structures

**Mohamad Zaki Hassan [1,\*], S. M. Sapuan [2], Zainudin A. Rasid [3], Ariff Farhan Mohd Nor [1], Rozzeta Dolah [1] and Mohd Yusof Md Daud [1]**

1   Razak Faculty of Technology and Informatics, Universiti Teknologi Malaysia, Jalan Sultan Yahya Petra, Kuala Lumpur 54100, Malaysia; ariffmdnor@yahoo.com (A.F.M.N.); rozzeta.kl@utm.my (R.D.); yusof.kl@utm.my (M.Y.M.D.)
2   Advanced Engineering Materials and Composites Research Centre, Department of Mechanical and Manufacturing Engineering, Universiti Putra Malaysia, Serdang 43400, Malaysia; sapuan@upm.edu.my
3   Malaysia-Japan International Institute of Technology, Universiti Teknologi Malaysia Jalan Sultan Yahya Petra, Kuala Lumpur 54100, Malaysia; arzainudin.kl@utm.my
\*   Correspondence: mzaki.kl@utm.my

**Abstract:** Banana fiber has a high potential for use in fiber composite structures due to its promise as a polymer reinforcement. However, it has poor bonding characteristics with the matrixes due to hydrophobic–hydrophilic incompatibility, inconsistency in blending weight ratio, and fiber length instability. In this study, the optimal conditions for a banana/epoxy composite as determined previously were used to fabricate a sandwich structure where carbon/Kevlar twill plies acted as the skins. The structure was evaluated based on two experimental tests: low-velocity impact and compression after impact (CAI) tests. Here, the synthetic fiber including Kevlar, carbon, and glass sandwich structures were also tested for comparison purposes. In general, the results showed a low peak load and larger damage area in the optimal banana/epoxy structures. The impact damage area, as characterized by the dye penetration, increased with increasing impact energy. The optimal banana composite and synthetic fiber systems were proven to offer a similar residual strength and normalized strength when higher impact energies were applied. Delamination and fracture behavior were dominant in the optimal banana structures subjected to CAI testing. Finally, optimization of the compounding parameters of the optimal banana fibers improved the impact and CAI properties of the structure, making them comparable to those of synthetic sandwich composites.

**Keywords:** banana fiber; impact response; compression after impact; natural fiber

---

## 1. Introduction

The research interest into utilizing natural fibers as reinforcement in polymers has dramatically increased during the last decade. It has been claimed that they can replace their synthetic polymer counterparts. Natural fibers, such as banana, kenaf, sugar palm, pineapple leaf, and empty fruit bunch, are abundantly available in tropical countries, especially in Southeast Asia and Papua New Guinea. Among the natural fiber composites, banana has attracted significant interest since it is biodegradable, not a health hazard, of low abrasivity, cheap, and offers good sound absorption capabilities. It belongs to a subclass of monocotyledonous herbaceous flowering plants in the genus *Musa*. This tropical plant originated from Brazil and was widely consumed after the American Civil War. Roughly, 72.5 million tons of banana fruit are produced yearly throughout the world [1]. The most widely recognized banana that is consumed by humans is a member of the *Musa acuminata* species. For each ton of banana

produced and harvested, around 100 kg of the fruit product is rejected and nearly 4 tons of biomass waste, including leaf, pseudo-stem, rotten fruit, peel, fruit-bunch-stem, and rhizome [2], are produced. In many nations, including Malaysia, the uses of banana fiber have been disregarded, despite research findings over the years [3].

Past studies have proven that the banana pseudo-stem is a promising fiber with a significant tensile strength [4] and stiffness [5]. Maleque et al. [6] mentioned that epoxy polymer reinforced by banana improved its mechanical strength by 90% and its impact strength by 40% compared to neat polymer. In addition, the optimal loading percentage of treated banana fiber incorporated into low-density polyethylene (LDPE) at a fiber loading of 20% offered the highest mechanical properties [7]. Further, the highest improvement of 4%, 11%, 14.5%, and 11.1% for Young's modulus and tensile, flexural, and impact strengths, respectively, was found for a 25% banana fiber loading that had been treated with alkali compared to values for neat LDPE [8]. In another study, Ahmed et al. [9] stated that 10% banana fiber loading treated with 5% alkaline suspension achieved the highest mechanical properties for acrylonitrile butadiene styrene. Venkateshwaran et al. [10] reported that an increase in banana fiber length of up to 15 mm and a 16% weight ratio increases the tensile strength and modulus of a banana/epoxy composite. However, in order to achieve higher mechanical properties, Udaya et al. [11] suggested an optimal fiber length and fiber weight ratio of 30 mm and 57%, respectively. However, the optimal findings for fiber treatment and fiber loading are still inconsistent. Higher loading and treatment of the banana fibers have led to poor interfacial adhesion and reduced mechanical properties.

A series of studies on the impact properties of a composite structure with banana fiber reinforcement have been conducted [12–16]. The impact strength of banana pseudo-stem unplastisized polyvinyl chloride composites were conducted by Zainudin and Sapuan [12] using the Izod impact test. It was found that fiber loading using banana fiber could possibly enhance the impact strength properties of the composites. Pavithran et al. [13] conducted the Charpy test in order to evaluate the effect of banana, sisal, and pineapple reinforcement on the fracture of coir/polyester composites and found that the sisal/polyester composites exhibited the highest value. The impact strengths of hybrid sisal, banana, coir, and sisal/banana/coir-fiber-reinforced epoxy were also compared by Balaji et al. [14]. Again sisal-fiber-reinforced epoxy offered a higher resistance to impact loading. In addition, Pothan et al. [15] examined the effect of fiber loading on a banana fiber/polyester composite following a low-velocity impact. They suggested that a fiber length from 30 to 40 mm and a fiber loading of 40% offered significantly higher impact resistance. Devireddy et al. [16] conducted impact response tests on banana, jute, and hybrid epoxy composites. It was mentioned that the hybrid composite offers an outstanding performance when compared to the individual natural fiber composites. Moreover, Narayana et al. [17] functionalized the hybrid nanocomposite made up of the nanoclay-reinforcement of either banana fiber, E-glass, or epoxy resin. The results showed that reinforcement using nanoclay inclusions was able to enhance the impact properties.

Energy absorption and compression residual strength capabilities are the main variables being analyzed in the impact testing. There have been several studies conducted on the low-velocity impact response and residual strength of composite structures. Dhakal et al. [18] studied the flexural strength after impact (FAI) of jute-reinforced unsaturated polyester composites. It was found that the FAI significantly decreased with the increase in test temperature and the damage assessment of the composites revealed delamination as the major failure mode. Ismail et al. [19] studied the post-impact behaviors of kenaf/glass hybrid composites with different weight proportions following a low-velocity impact test. The compression after impact (CAI) test demonstrated that the compression damage decreased as the impact energy increased. It was further discovered that the kenaf/glass hybrid composite with a 25% kenaf fiber weight ratio gave results that are comparable to those of a glass laminate composite. Mohd Nor et al. [20] examined the effect of nanofiller in a bamboo/epoxy composite to enhance the CAI properties. The addition of carbon nanotubes (CNTs) into the composite improved the compression post-impact response properties. The compressive residual strength test was also carried out on flax/polylactic acid (PLA) laminates [21]. The absorbed energy and normalized residual

strength were analyzed and the main failure mode in a composite laminate was identified as being fiber failure. Numerous studies have explored the post-impact behavior of glass/epoxy composites, Kevlar/epoxy composites, and other synthetic types of composites. However, only limited findings on the CAI behavior of optimized banana fiber reinforced epoxy composites have been reported.

In order to improve the mechanical behavior of the natural fiber composites, many researchers have implemented a well-developed statistical approach including the Taguchi method and the response surface method (RSM). The optimal parameters, such as temperature, molding time, and volume fraction of kenaf-reinforced polyethylene composite, were determined using the Box–Behnken response surface method for ballistic protection, as reported by Akubue et al. [22]. Moreover, Yaghoobi and Fereidoon [23] evaluated the effect of the fiber load, fiber length, and compatibilizer content on the tensile strength and modulus using the Box–Behnken design. The results showed that the $R^2$ values and normal probability plots were in good agreement. Furthermore, Roslan et al. [24] investigated the mode I fracture toughness of optimized alkali-treated bamboo using the Box–Behnken method. It was suggested that this statistical analysis approach is highly suitable for optimizing the parameters for alkaline-treated bamboo fibers.

In previous work, the Box–Behnken method was used to determine the optimal parameters, including fiber length, fiber loading, and chemical treatment concentration, for a banana/epoxy composite [25]. However, there is a need to explore the behavior of this optimal fiber for sandwich structures to fill the knowledge gaps in this particular field of study. In this research, the optimized banana-fiber-reinforced epoxy composites were laminated with carbon/Kevlar twill woven skins to form banana epoxy sandwich structures. This current study focused on the low-velocity impact and compression after impact (CAI) response of these structures. The optimal behavior of banana fiber sandwich panels was compared to that of the synthetic fibers including Kevlar, carbon, and glass fibers. Prior to that, the tensile properties of neat epoxy-resin and optimal banana composite were also discussed. Further, details of the loading behavior, toughness, and damage evolution were obtained.

## 2. Materials and Methods

### 2.1. Materials

The banana pseudo-stem fiber from the *Musa acuminata* species [26] with a diameter range from 500 μm to 1 mm was supplied by Innovative Pultrusion Sdn Bhd, Seremban Negeri Sembilan, Malaysia. The EpoxAmite™ 100 epoxy resin and EpoxAmite™ 102 medium hardener [27] used as the base matrix were purchased from Kird Enterprise, Nilai Negeri Sembilan, Malaysia. This epoxy-resin system was mixed to a weight ratio of 10:2.9 g. The banana fibers were soaked in sodium hydroxide (NaOH) obtained from Orioner Hightech Sdn Bhd, Cyberjaya Selangor, Malaysia. In this study, a 2/2 twill weave carbon/Kevlar hybrid with a density of 210 g/m$^2$ was used for the skins. Carbon and Kevlar fiber tow was used as a comparison for the banana fiber. Those were purchased from EasyComposite Ltd., Stoke-on-Trent, UK [28]. Glass fibers were purchased from Alsey Kimia Sdn Bhd, Puchong Selangor, Malaysia. Detailed properties of all materials are given in Table 1.

**Table 1.** Properties of fibers, skins, and EpoxAmite™-102 hardener.

|  | Banana | Kevlar | Carbon | Glass | Carbon Kevlar Twill | EpoxAmite™-102 Hardener |
|---|---|---|---|---|---|---|
|  | [26] | [28] | [28] | [28] | [28] | [27] |
| Density (kg/m$^3$) | 1350 | 1340 | 1780 | 2600 |  | 1110 |
| Flexural Strength (MPa) | 52 |  |  |  |  | 84.25 |
| Tensile Strength (MPa) | 54 | 3260 | 4900 | 3450 |  | 56.4 |
| Young's Modulus (GPa) | 3.49 | 60–80 | 250 | 72–77 |  | 3.1 |
| Elongation (%) |  | 4.4 | 2 | 4.7 |  |  |
| Weight (g/m$^2$) |  |  |  |  | 210 |  |
| Weft |  |  |  |  | 2(C)-1(K) |  |
| Warp |  |  |  |  | 2(C)-1(K) |  |

**Table 1.** *Cont.*

|  | Banana | Kevlar | Carbon | Glass | Carbon Kevlar Twill | EpoxAmite™-102 Hardener |
|---|---|---|---|---|---|---|
|  | [26] | [28] | [28] | [28] | [28] | [27] |
| Cellulose (%) | 63–64 |  |  |  |  | - |
| Hemicellulose (%) | 19 |  |  |  |  | - |
| Lignin (%) | 5 |  |  |  |  | - |
| Mixed viscosity (kg/ms) | - |  |  |  |  | 0.65 |
| Specific volume (m³/kg) | - |  |  |  |  | $9.03 \times 10^{-4}$ |

## 2.2. Fabrication of Composites

Initially, to eliminate any surface impurities, a long banana fiber, as shown in Figure 1a, was washed with deionized water and dried in a circulation oven Model H750CLAB200D16 (CMH Ltd., Lancing, UK) at 70 °C for 9 h. Then, the fibers were soaked in 5.45 wt.% sodium hydroxide (Figure 1b) for 5 h according to the optimal conditions suggested by the Box–Behnken design [25]. These fibers were ground down using a Cheso Model N3, (Cheso Machinery Pte. Ltd., Loyang Way, Singapore) crusher machine (Figure 1c). In order to obtain the 3.35-mm fiber length, a multi-stage sieve (Figure 1d)—model BS410/1986 (Endecots Ltd., London, UK)—fixed to a rotational shaker (Endecots Ltd., London, UK) (Figure 1e) was used. The speed of this shaker was maintained at 275 rpm for 45 min. The short banana pseudo-stem fibers were utilized as the reinforcement, as illustrated in Figure 1f. Similar processes were repeated for the Kevlar, carbon, and glass fiber tows, except for the alkaline treatment. These synthetic fibers were chopped using a carbon fiber shear cutter Model 3670C-8 (EasyComposite Ltd., Stoke-on-Trent, UK) and sieved using a rotational shaker. Then, the epoxy-resin matrix with fibers was gradually mixed at 29.86 wt.% of the fiber loading.

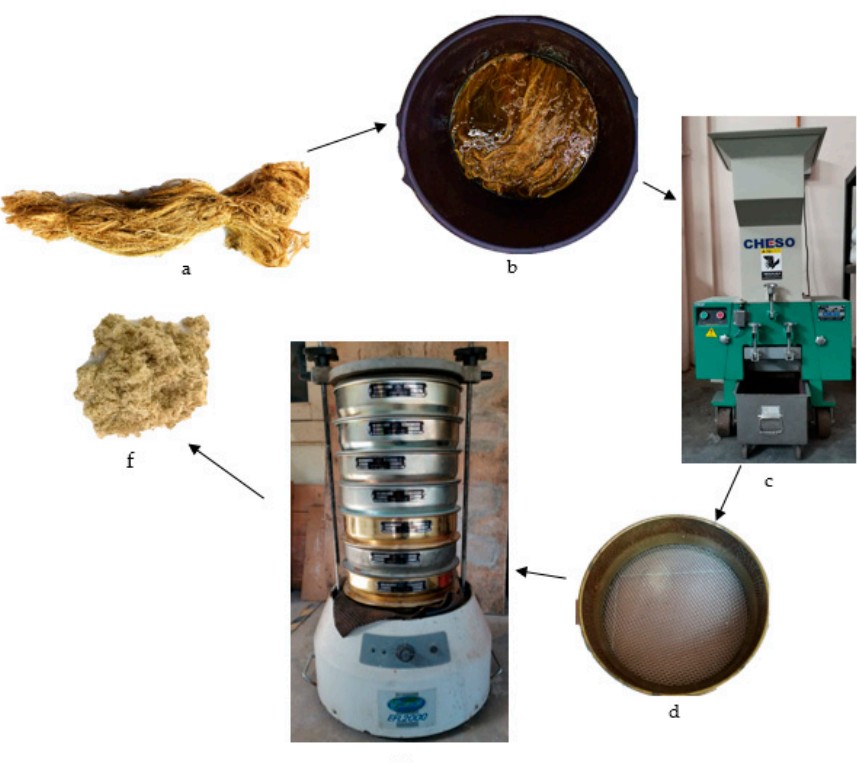

**Figure 1.** Photos of (**a**) long banana fibers, (**b**) chemically retted fibers, (**c**) fiber chopping machine, (**d**) 3.35-mm brass frame sieve, (**e**) rotational shaker, and (**f**) short banana fibers.

### 2.3. Tensile and Sandwich Structure Preparation

The tensile test specimens of banana epoxy composites were fabricated using the mold shown in Figure 2. A dog bone specimen was manufactured following ASTM D638 [29].

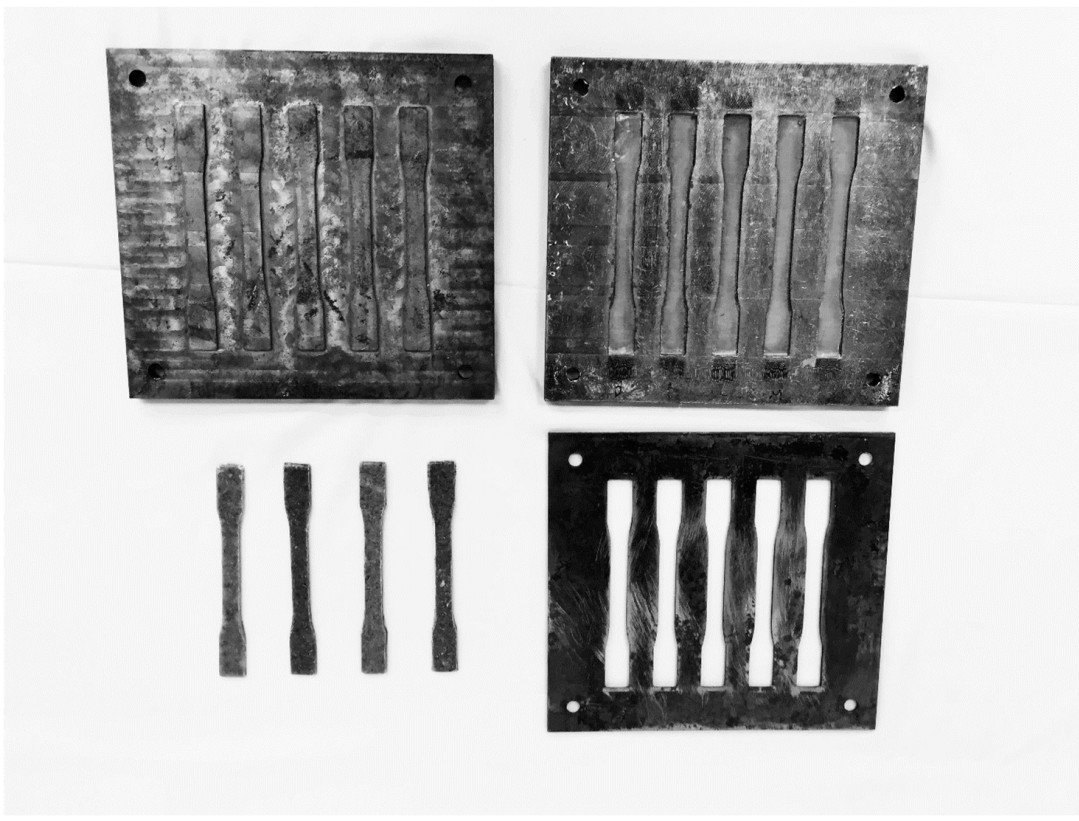

**Figure 2.** Photo of tensile specimen fabrication.

In order to fabricate the sandwich structure, a banana and synthetic fiber epoxy composite was layered with carbon/Kevlar plies. Here, these 2/2 woven hybrid skins were initially cut to a size of $300 \times 300$ mm before being placed into picture frame molds. Prior to that, those fibers were gradually mixed with epoxy-resin paste. Further, the sandwich structures were prepared using a hot press machine. The specimens were then heated to 70 °C for one hour under a pressure of 1 bar before leaving them to cure overnight. The samples were inspected visually before being sectioned into $150 \times 100$ mm samples. In this study, at least five samples for each configuration were examined. The configuration of this structure is shown in Figure 3.

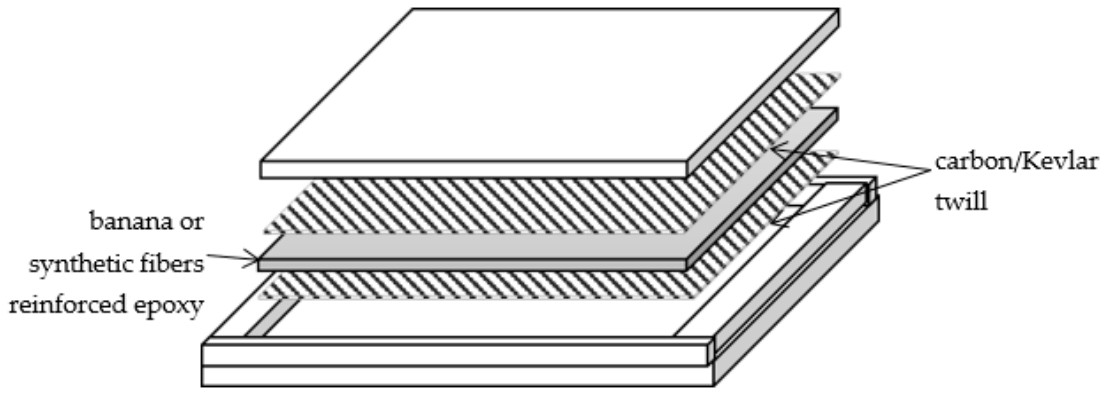

**Figure 3.** Schematic of the fiber-reinforced epoxy sandwich structure fabrication.

### 2.4. Tensile Test Properties

The tensile test was conducted using a Universal Testing Machine, (Shimadzu AGX-S, Kyoto, Japan). This table stand tensile machine was fixed with a 10 kN load cell. Testing was obtained at a crosshead speed of 1 mm/min. Tests were conducted for all five samples and their average values were used as the final result.

### 2.5. Low-Velocity Impact Test

The low-velocity impact properties of the sandwich structures were evaluated using a floor-standing impact tower—CEAST 9340, (Instron, Pianezza TO, Italy)—according to the ASTM D3763 standard [30]. The tests were investigated at energy levels from 5 J to 20 J using a hemispherical steel indenter with a 12.7-mm diameter. Then, the load–displacement traces were recorded, followed by calculating the energy absorbed from the area under the curves.

### 2.6. Dye Penetrant Application

The damage area subjected to the impact was easily located using a dye penetrant that complied with the ASTM E1417 standard [31]. This nondestructive testing utilized a Spotcheck SKL-SP2 kit supplied by Kird Enterprise, Nilai Negeri Sembilan, Malaysia. Initially, all samples were cleaned using a solvent to remove dirt, sand, and grease. The aerosol red-color penetrant was sprayed on the surface and the samples were left for 10 min. Then, the excess red penetrant was gently cleaned, and finally, the well-shaken developer was applied to the impacted area to increase the visibility of the damaged region. The red spots that remained visible on the sample's surface were due to the damaged area affected by the impact loading.

### 2.7. Damage Area Measurement

Specimen damage for an area of the impacted sandwich structure was characterized using ImageJ version Java 1.8.0_172 (National Institutes of Health (NIH), Maryland, US and Laboratory for Optical and Computational Instrumentation (LOCI), University of Wisconsin, US) software. Initially, to convert the color-scanned photo to grayscale, the Image/Type/8-bit command was used. Then, a straight line was drawn from edge-to-edge of the photo as a known measurement distance. The Analyze/Set Scale/Known Distance/Unit of Measurement (mm)/Global command was chosen to set the scale parameter. Moreover, the damage area color was inverted using the Image/Adjust/Threshold manual setting. Finally, the calculated area outlines were measured using a rectangular selection tool called the Analyze/Analyze Particle function.

### 2.8. The Compression after Impact Test

The residual compressive strengths of the post-impacted specimens were evaluated using the CAI test setup. The specimens were fully clamped using the anti-buckling Boeing CAI test fixture according to the ASTM D7137-17 standard [32]. An in-plane compression load was applied at a crosshead displacement rate of 1.25 mm/min until the specimen failed. A Shimadzu AGX universal testing machine fitted with a 300 kN load cell was used to obtain the load–displacement traces. The compressive residual strength of the materials was characterized using the ultimate load prior to failure over the cross-sectional area of the specimen.

## 3. Results and Discussion

### 3.1. Tensile Properties of Composites

Figure 4 illustrates the stress–strain curves of the neat epoxy system, banana fiber composite, and optimal banana reinforced epoxy composites. It can be seen that the maximum stress and tensile modulus of the optimal banana fiber reinforced epoxy composite were increased by 66% and 22%,

respectively, compared to the neat epoxy resin. It is suggested that the properly optimized the fiber blending condition improved the load-bearing capabilities between the fibers and matrixes. A similar finding was also reported by Yaghoobi and Fereidoon [23]. In this figure, the untreated, 0.25-mm length, and mixed at 50 wt.% fiber loading of a banana-fiber-reinforced epoxy composite was also included for comparison. Interestingly, these composites unfolded at lower tensile strength and modulus values than the virgin epoxy-resin. In addition, the tensile modulus of this unoptimized composite was found to be 792 MPa, which was lower than that of the optimal banana epoxy composite with 1628 MPa. It was demonstrated that adding inappropriate natural fiber "debris" in the polymeric matrixes decreased the material properties of the structure.

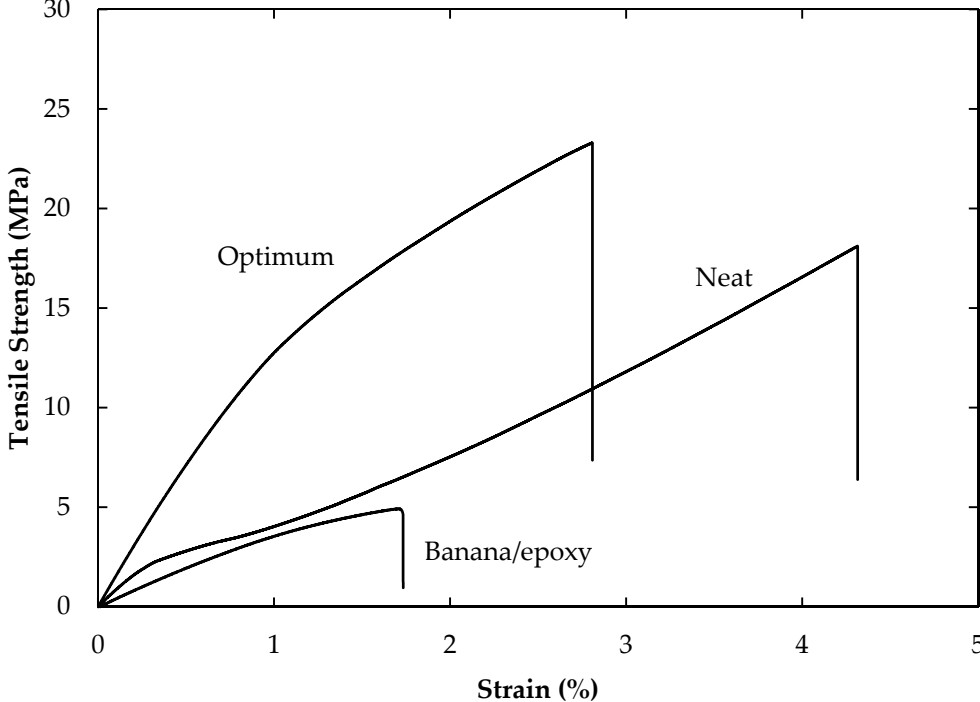

**Figure 4.** Typical stress–strain curve for neat, optimal banana/epoxy composite, and banana/epoxy composite.

In addition, the tensile modulus of this impair composite was obtained to be 792 MPa, which was evidently lower than that of the optimum banana epoxy composite with at 1628 MPa. It can be demonstrated that adding inappropriate "debris" of natural fiber in the polymeric matrixes decreased the material properties of the structure. The average and standard deviation of tensile properties of the composites are illustrated in Table 2.

**Table 2.** The average and standard deviation values of the modulus of elasticity, tensile strength, and percentage elongation of the virgin epoxy and natural fiber composites.

| Material | Tensile Strength | Modulus of Elasticity | Elongation (%) |
| --- | --- | --- | --- |
| | (MPa) | (MPa) | |
| Neat Epoxy | 19.18 ± 2.6 | 980 ± 1.8 | 4.21 ± 0.8 |
| Banana/Epoxy | 8.23 ± 1.4 | 792 ± 2.6 | 1.72 ± 1.3 |
| Optimum Banana/Epoxy | 23.30 ± 2.3 | 1628 ± 4.8 | 2.81 ± 2.4 |

### 3.2. Low-Velocity Impact Response of Sandwich Structures

Figure 5 illustrates typical load–displacement traces for the low-velocity impact response at four different impact energy levels correspond to the optimal banana, Kevlar, glass, and carbon epoxy sandwich structures. All traces initially exhibited a similar pattern with the force increasing in an

almost linear manner before reaching a peak load. This was associated with the plastic deformation of the sandwich structures directly beneath the hemispherical impactor. Figure 5a presents typical load versus displacement traces for the impact loading of the optimal banana epoxy sandwich structures. It can be seen from the figure that increasing the incident impact energy increased the effective slope of the curve up to 3 mm of displacement, suggesting a rise in the contact stiffness of the system. Beyond this point, a non-linear trace pattern was observed, resulting in the initiation of unstable crack propagation and fiber fracturing. A very drastic fall in the contact load was apparent in the trace for the 20 J impact energy level due to sub-critical propagation and brittle failure of the core. Continued loading often resulted in microcracking and instabilities, with this being most evident in the 15 J impact energy trace where very small load jumps were apparent in the trace at displacements above 3 mm. In addition, the enclosed area within the loading and unloading curves was a measure of the energy absorbed due to the damage in the laminates. Then, the impactor was rebounding since the force and deflection values decreased in the unloading phase. At all the energy levels used in this study, the impactor did not fully perforate the sandwich structures and a portion of the impact energy was conserved as elastic energy. This energy was then transferred back to the steel impactor, causing it to rebound. He et al. [33] suggested that if the core structure was a highly brittle material, the rapid rebound of the skin under impact can result in debonding at the interface between the adjacent layers of the face sheet and the core. Figure 5b–d reveals the typical load displacement traces for Kevlar, glass, and carbon epoxy sandwich structures after being subjected to 5, 10, 15, and 20 J impact energies. From the figure, it is evident that the Kevlar/epoxy composite has a characteristically higher contact stiffness than that of the other composite structures; however, the carbon/epoxy structure offered less "resting" displacement during the unloading phase. It can be suggested that this has resulted in less of a dent depth for the carbon epoxy structures.

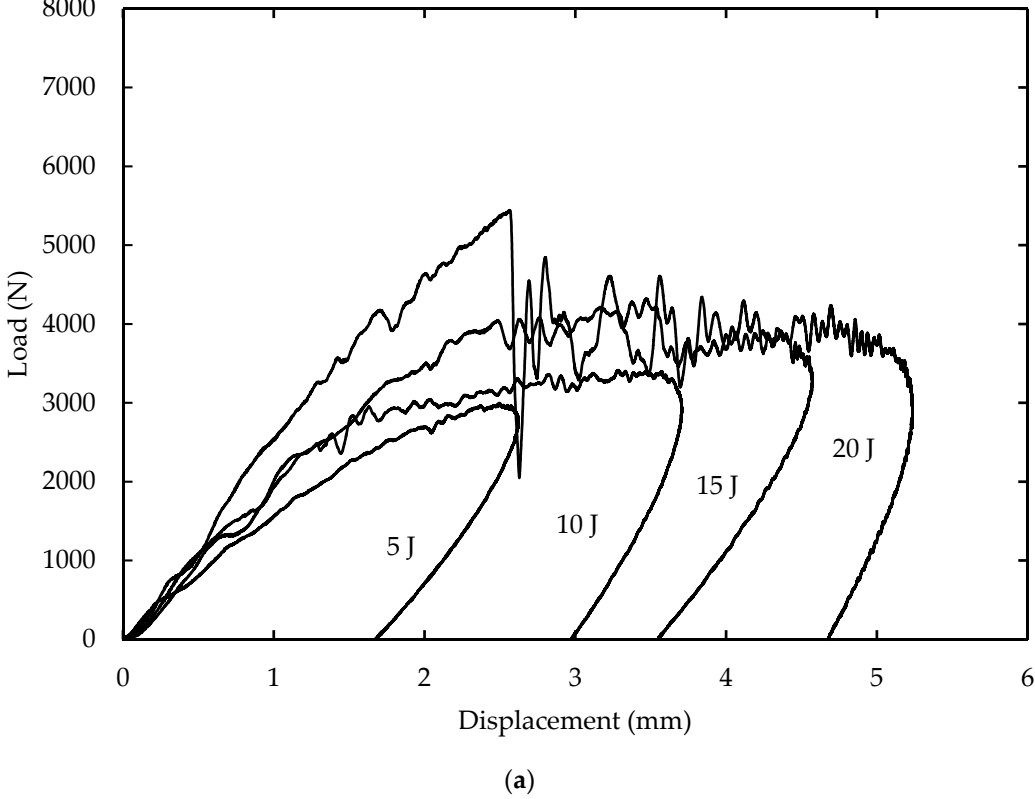

(**a**)

**Figure 5.** *Cont.*

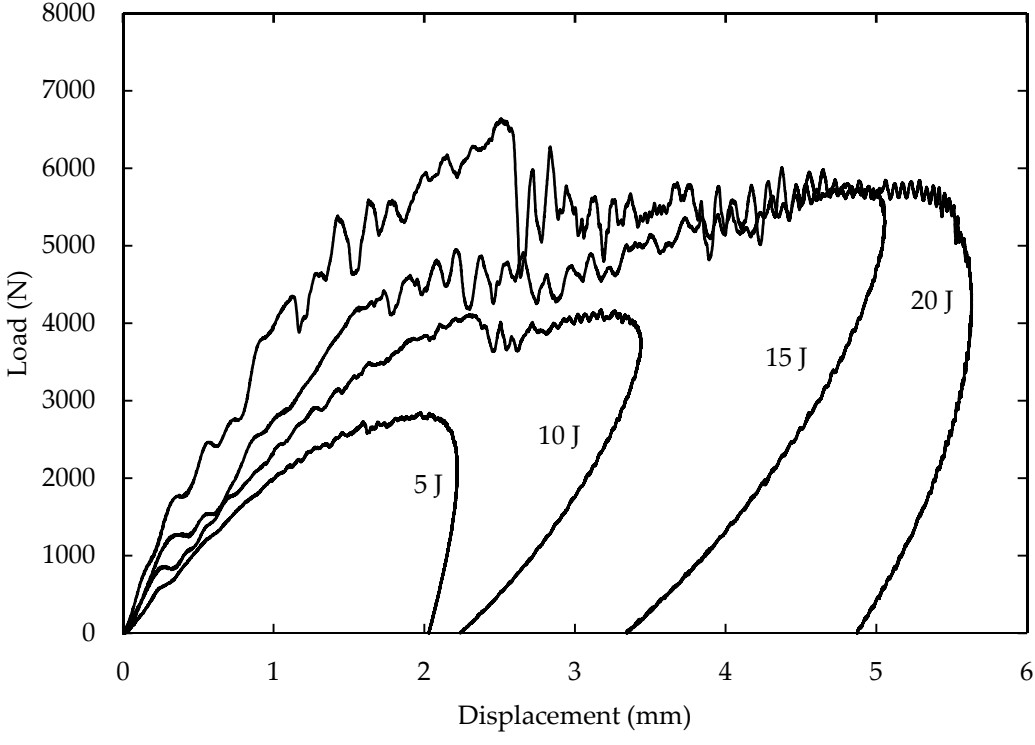

(**b**)

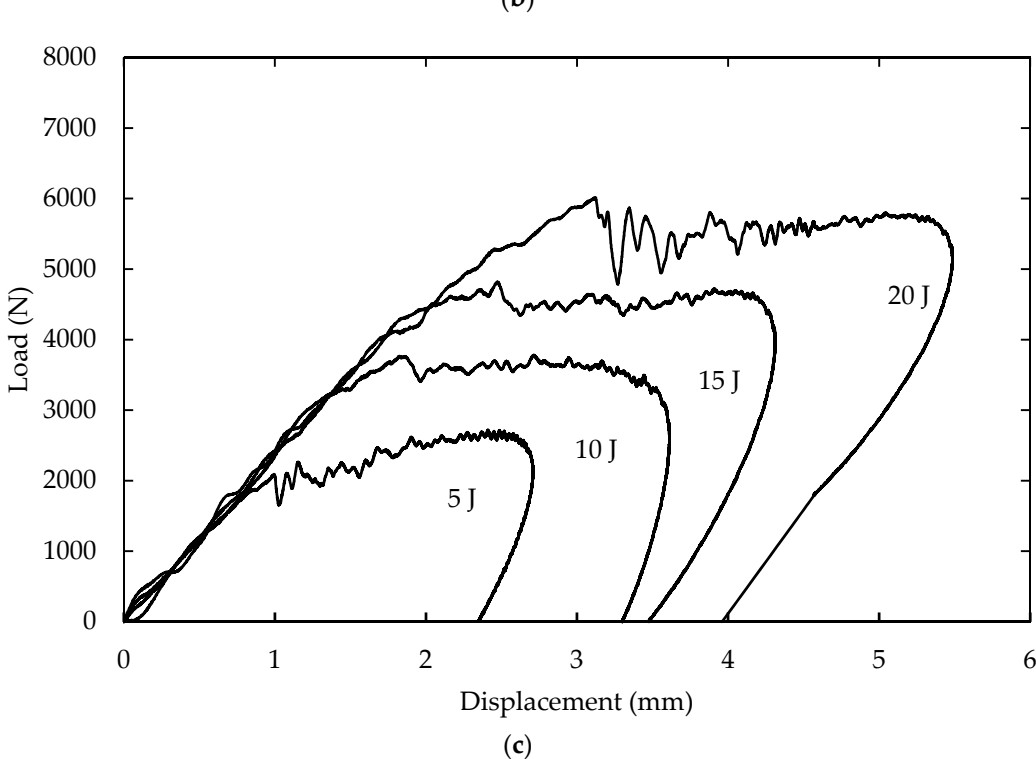

(**c**)

**Figure 5.** *Cont.*

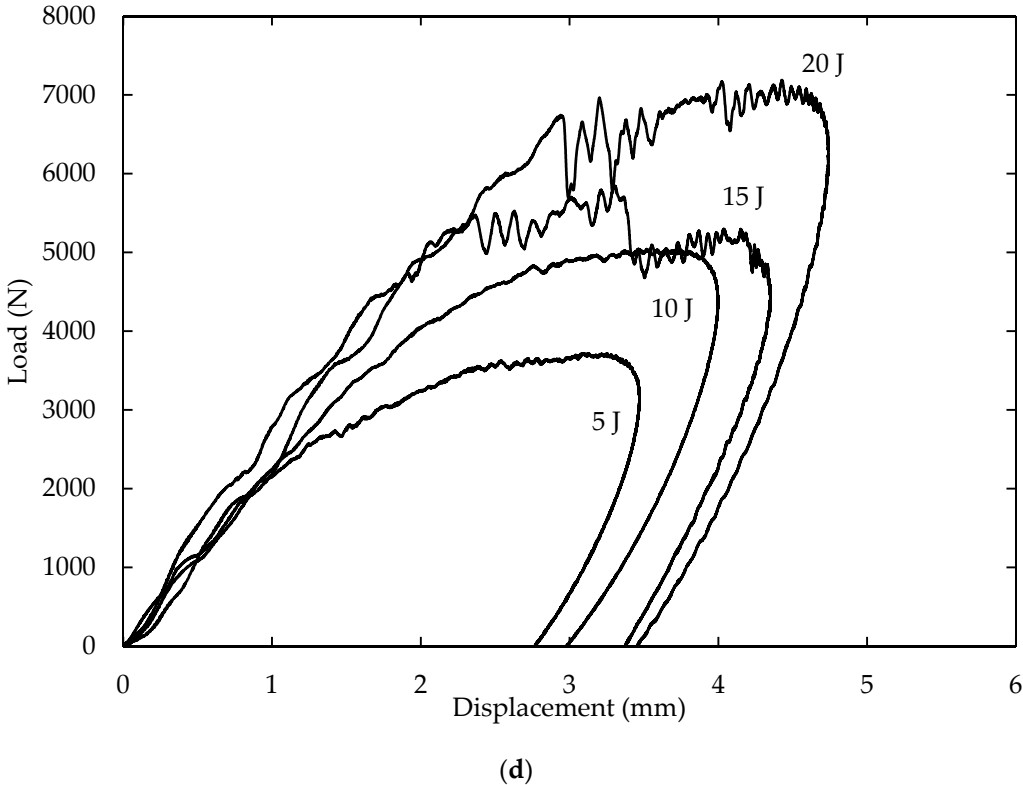

(**d**)

**Figure 5.** Typical load–displacement traces of (**a**) banana, (**b**) Kevlar, (**c**) glass, and (**d**) carbon/epoxy sandwich structures.

The variations in the peak force at different impact energy levels for composite structures are shown in Figure 6. It can be observed that the peak impact force was increased with increases in the energy level. A similar observation was also reported by Olsson et al. [34]. In addition, the optimal banana/epoxy composite offered a lower peak force than those of the synthetic/epoxy structures. Joseph et al. [35] claimed that fiber pullout and fiber compatibility were the major contributors to the toughness of the composite. Thus, synthetic fibers offered a significantly higher peak load, suggesting higher bonding capabilities with the matrix due to the surface smoothness and consistency of the cross-section. In the case of natural fibers, such a mechanism was not favored due to the mechanical interlocking between fibers and the matrices. In this study, to increase the compatibility between the banana fiber and the matrix, the chemical treatment was promoted; however, the peak force still remained unimpressive.

The peak displacement indicates the maximum deformation of the sandwich panels, which results in a significant area of damage after subjection to different energy levels. Figure 7 shows representative maximum dent depth against energy level traces for different core composites. In general, the maximum displacement steadily increased with rising impact energy. It can be seen that carbon fiber had the highest dent depth at low energy levels and it was the lowest at higher impact energies. This was in part due to the high in-plane tensile properties of the material. Properties of the core, including rigidity and brittleness, also influenced this behavior. In a recent study by Chen and Hodgkinson [36], it was found that at high peak displacement, the uppermost skins of the specimens exhibited more evidence of splitting and delamination of the surface ply. Further, comparing the maximum displacement values for optimum banana and synthetic structures, it can be seen that the values vary between those structures. This helps to explain why the differences between the behavior of core structures were within the scattering; as a result, peak displacement is another independent parameter of sandwich panels.

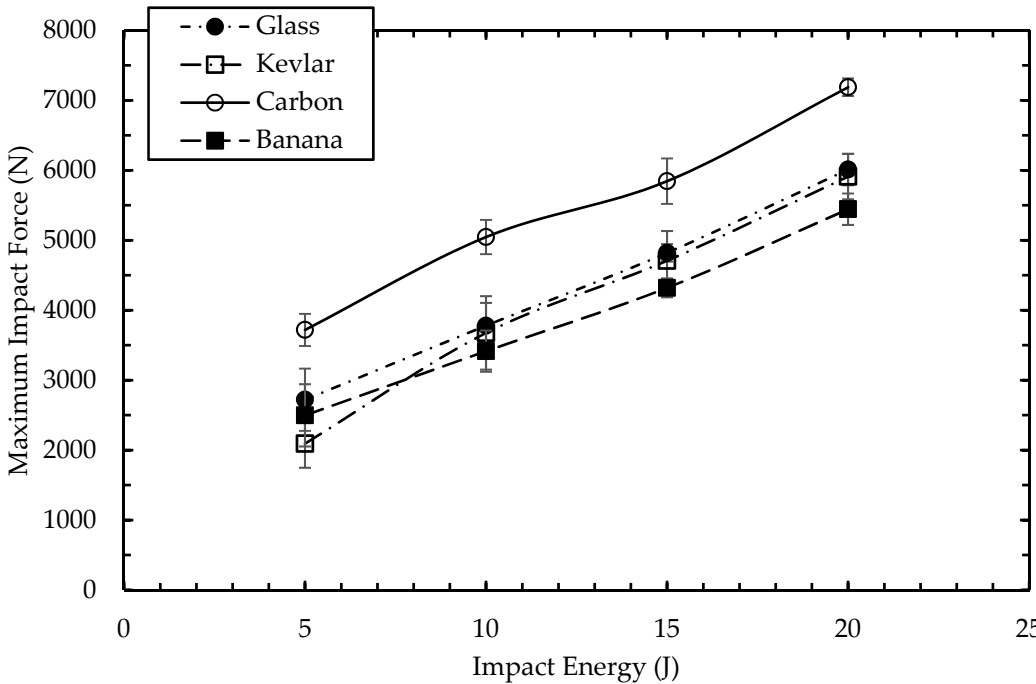

**Figure 6.** The variation of the maximum impact force at different impact energy levels for composite structures.

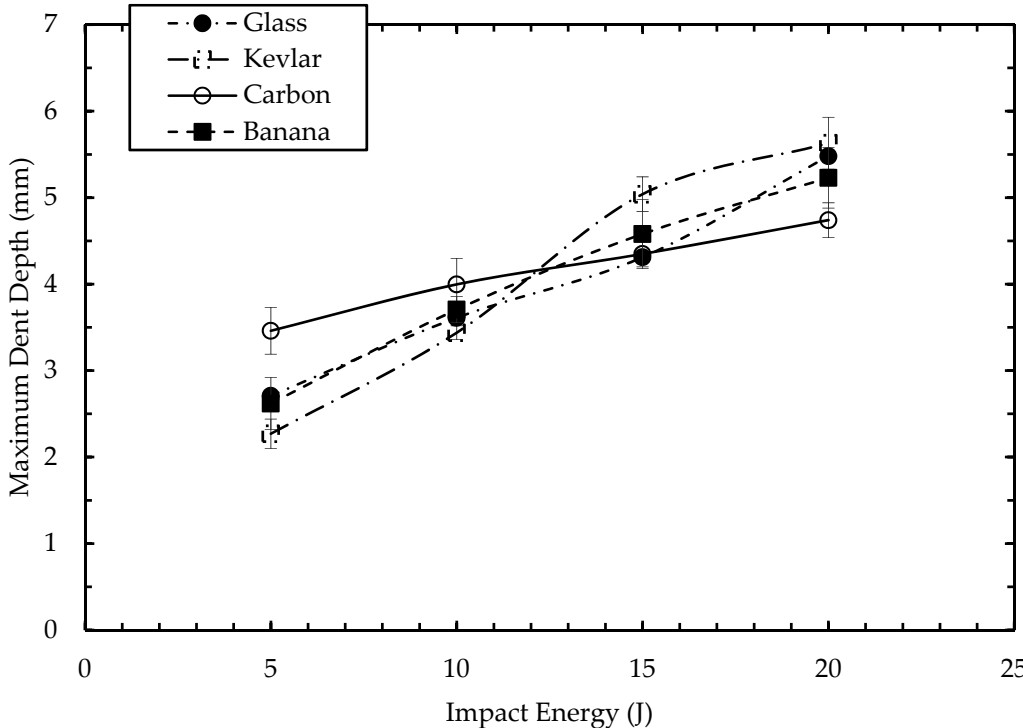

**Figure 7.** Variation of the maximum dent depth against energy level traces for composite structures.

The total impact energy is a combination of the energy absorbed and the elastic energy loss of the impacted structures [37]. If the percentage of the energy absorbed is high, the energy converted to elastic energy lost by the impactor is low. As a result, the impact energy is fully transferred to the structures at the point of maximum displacement [38]. In order to evaluate the impact response and resistance of the composite structures, the energy absorbed against impact energy traces are shown in Figure 8. The absorbed energy was measured by calculating the area under the force–displacement

traces. An observation of the figure suggests that the absorbed energy tended to increase with the incident impact energy, although there did appear to be scattering in the findings, particularly between the optimal banana/epoxy and carbon/epoxy composites. The lower energy absorption means that there was not much energy lost due to failure. Thus, each failure mechanism, including matrix cracks, interlayer failures, delamination, and fiber breakage, absorbed a fraction of the impact energy. The variation of energy absorbed depends on the skin thickness [39], the mechanical properties of fibers and matrices [40], the density of the core [41], and the impactor head [42].

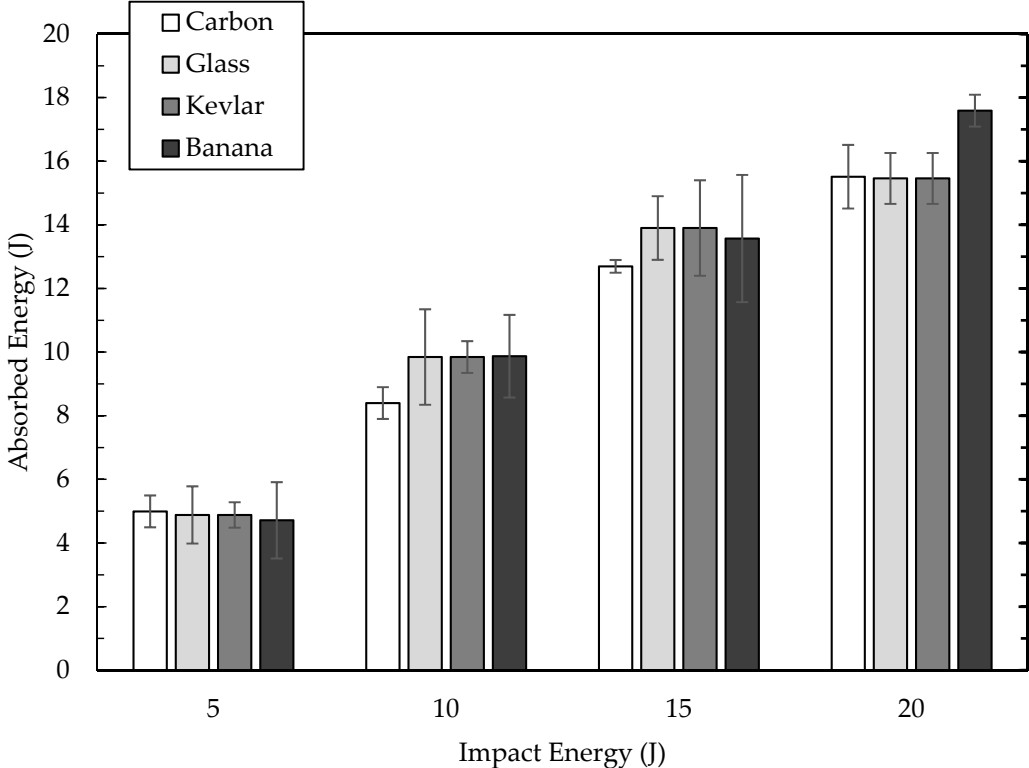

**Figure 8.** Typical absorbed energies at various incident energy levels.

The evolution of the damage area with impact energy for the optimum banana/epoxy sandwich structure is presented in Figure 9. The sample data was elucidated using a dye penetrant inspection beginning at the uppermost impacted face and followed by measurement of the area using ImageJ software. According to the load–displacement curves (Figure 5), the onset of damage, essentially matrix cracking, was at approximately 3000 N. As the impact energy increased, greater propagation of the damage area in the transverse direction of the specimen was obtained. As can be seen from Figure 9a, a closely peanut shape of visible marks, just like in plain laminates [43], was observed, which was dominated by matrix cracking of the skin. Further loading often resulted in delamination and fiber breakage, which was revealed by the changes of dye color, as shown in Figure 9b–d, which illustrate a similar shape for the dented areas. It can be suggested that this damage mainly corresponds to delamination of the skin and brittle fracture of the core. Here, a small permanent indentation (residual indentation) was created on the uppermost skin, which added to the fracture resulting from the composite core around the impact site. If the core layer is a brittle material, the fast rebound of the face sheet under the impact will result in debonding at the interface between the face sheet and the core. Selver et al. [44] mentioned that larger dent depths in the structure may be attributed to impact energy that was absorbed by a smaller area, which generates a greater plasticity of the composite, stiffness degradation, and more localized damage under the hemispherical steel impactor.

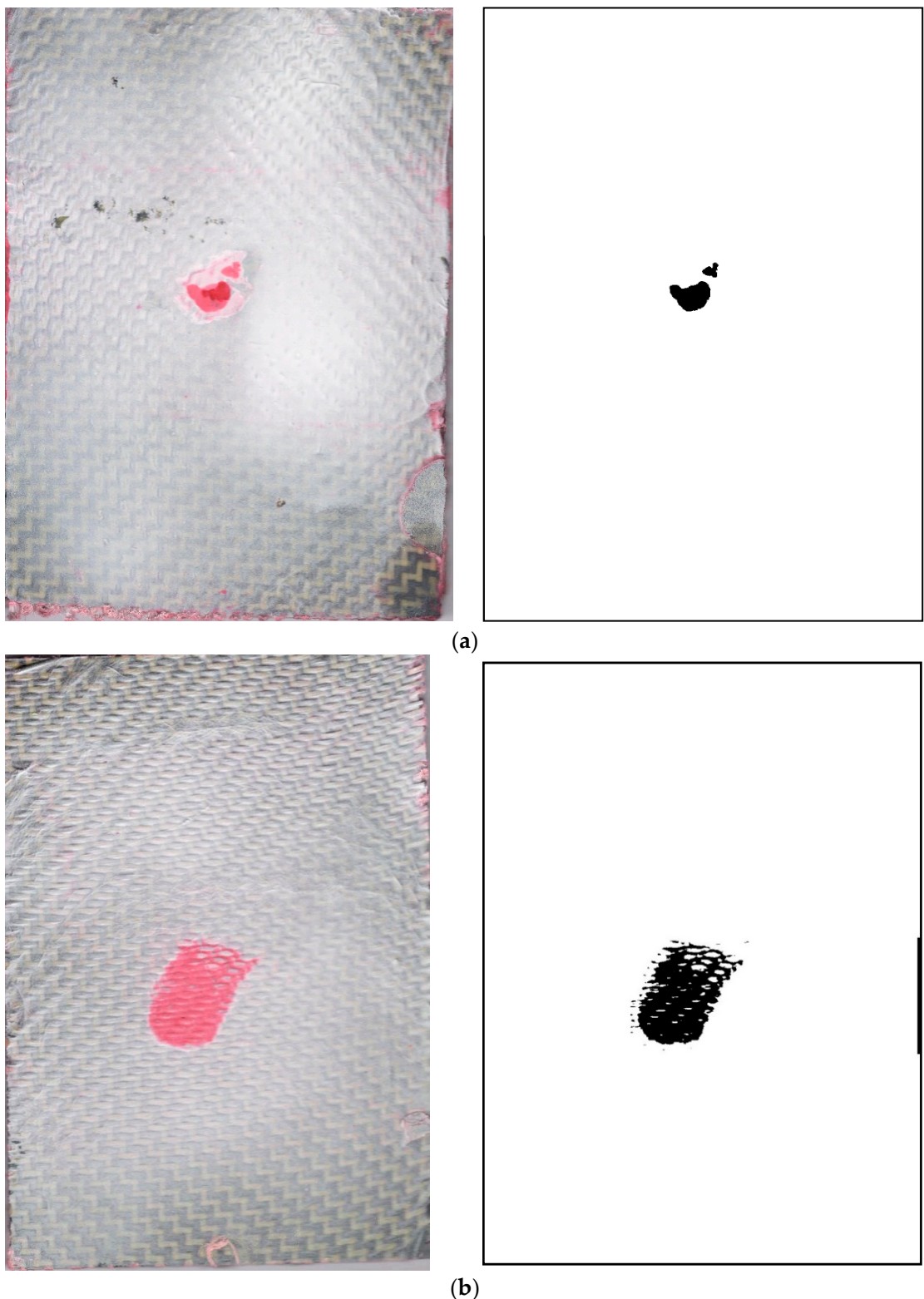

(**a**)

(**b**)

**Figure 9.** *Cont.*

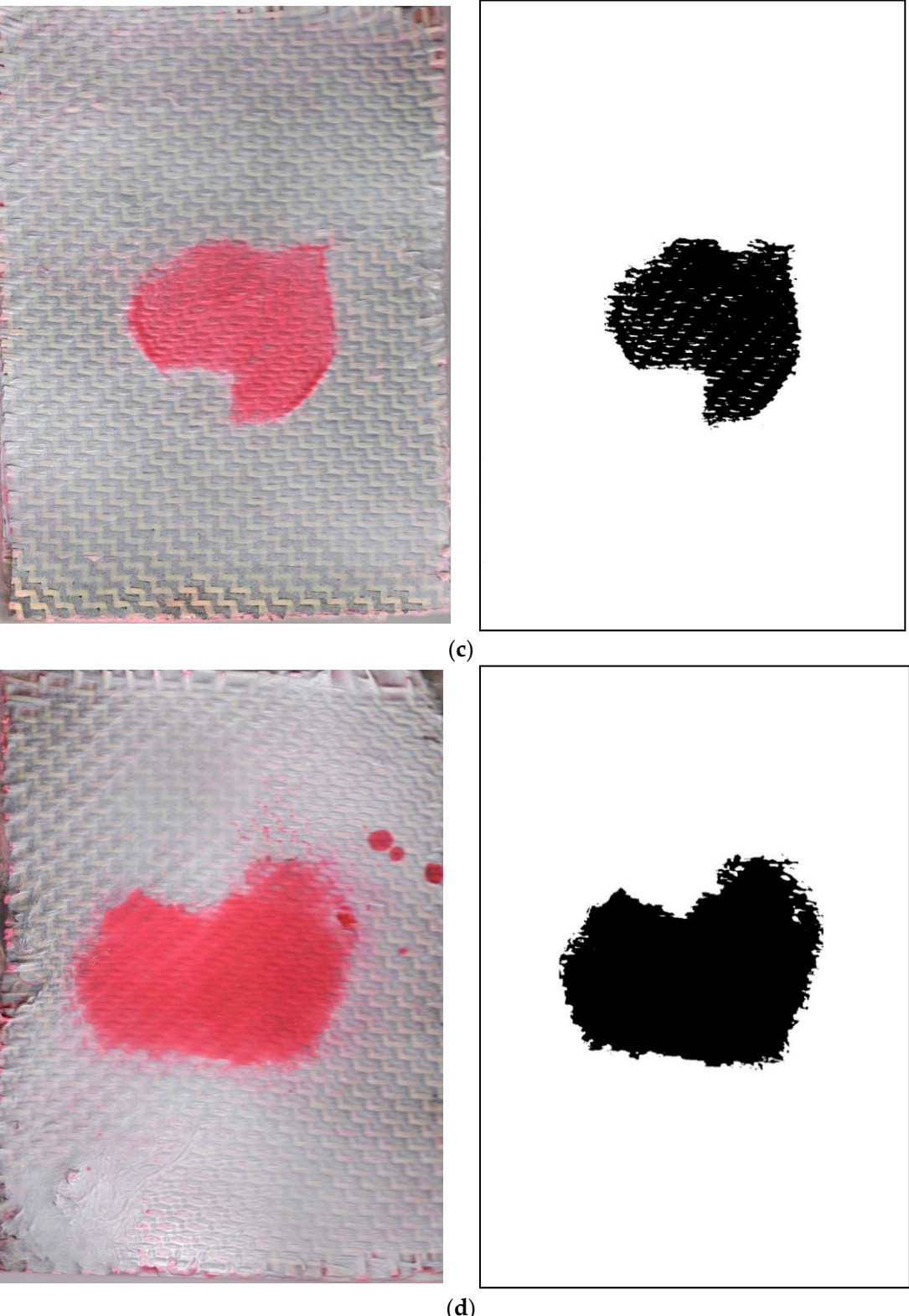

**Figure 9.** Damage area of impacted specimens of the banana/epoxy sandwich structure with respect to different impact energy levels: (**a**) 5 J, (**b**) 10 J, (**c**) 15 J, and (**d**) 20 J.

Figure 10 shows the front- and back-face damaged areas versus the nominal impact energies. The areas were calculated by post-processing the results of the dye penetrant test, as shown in Figure 9. Many works [45,46] reported that the relationship between the in-plane damaged area and the incident

impact energy of the impactor is linear. As can been seen from Figure 10a, in the impact process for sandwich structures where the energy level exceeded 10 J, after the damage to the top skin and the core, there was an amount of residual energy, which was associated with the damage to the bottom face sheet. A pronounced in-plane damage area for the back face of the optimal banana structures is shown in Figure 10b, suggesting a weak core material leading to serious failure of the bottom ply.

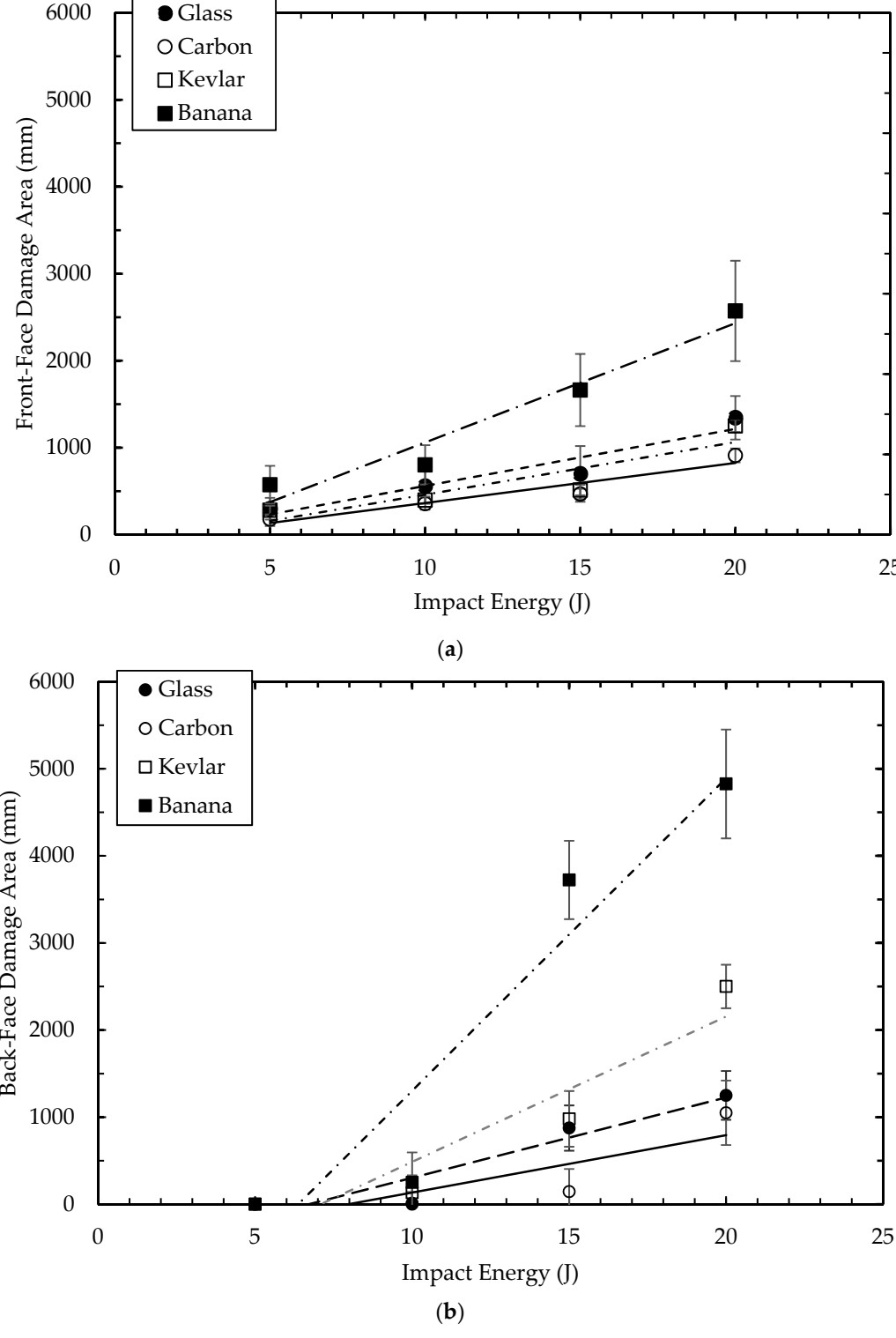

**Figure 10.** Nominal (**a**) front- and (**b**) back-face damaged areas versus incident impact energies.

### 3.3. Compression after Impact (CAI) of Structure Properties

Figure 11 shows the variations of CAI strength performance with the displacement for the optimal banana, carbon, Kevlar, and glass sandwich structures following an impact energy of 10 J. For all sandwich panels, the structures exhibited an initial linear CAI strength before reaching a maximum value, following which the strength gradually fell, an effect that was also reported previously [47]. The figure shows that an increase in peak stress resulted in a decrease in the displacement, highlighting the presence of the stronger and stiffer core structures. The carbon/epoxy based sandwich structure was seen to offer a greater residual strength than those of other systems. Furthermore, the findings also show that the CAI strength of the natural-fiber-reinforced epoxy was still lower than its synthetic fiber counterparts. The differences in residual stress were recorded at 22%, 63%, and 136% between banana and Kevlar, glass, and carbon respectively.

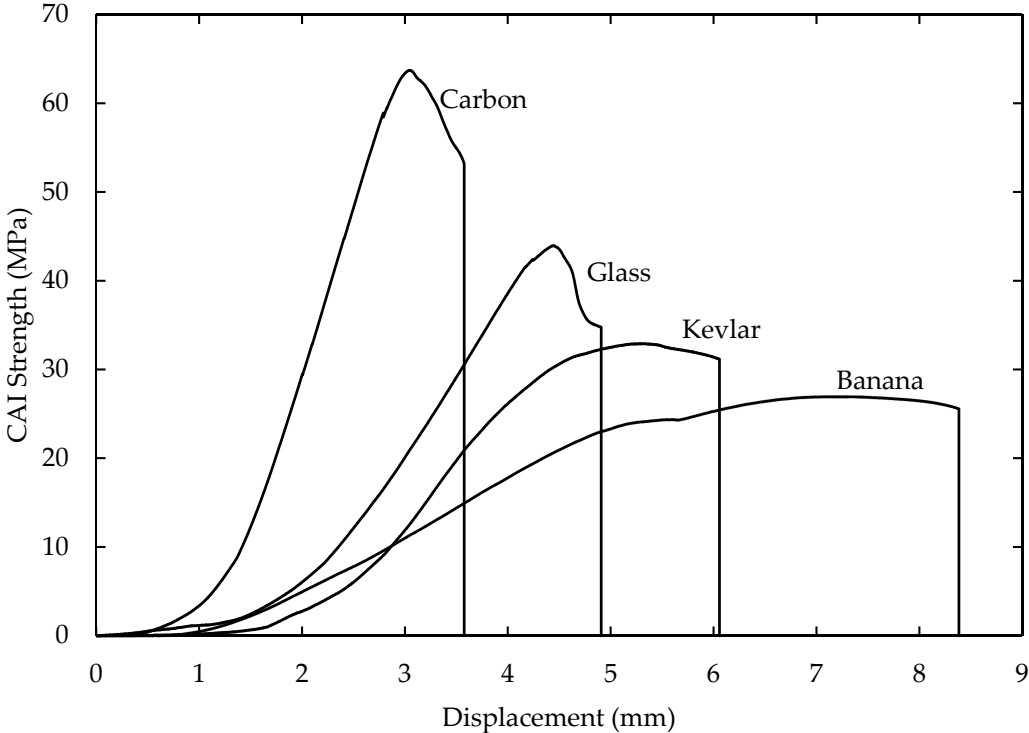

**Figure 11.** The variation of compression after impact (CAI) strength with the displacement for four different sandwich structures subjected to an energy impact of 10 J.

Figure 12 illustrates the average CAI strength versus impact energy traces of the sandwich structures, including those of unimpacted specimens. As observed, the compression strengths decreased with the increase in impact energy. The carbon fiber sandwich structure exhibited a massive residual strength under all impact energies with the optimal banana composite system providing the least. The outstanding residual strength performance of the carbon/epoxy sandwich structure was due to the high resistance to shear cracking between the carbon and epoxy of the core. The use of the surface treatment and fiber conditions of the banana pseudo-stem fiber in order to increase the surface bonding between the fiber and matrix was still insufficient and failed to offer high resistance to CAI strength compared to synthetic fibers. With respect to CAI behavior in the sandwich structures experiments, core breaking and skin buckling were identified as the major failure modes of the sandwich structures [48]. Castanié et al. [49] mentioned that matrix cracking and delamination between the face sheet and the core were found to be the most common damage mechanisms of the sandwich structure under residual compression strength.

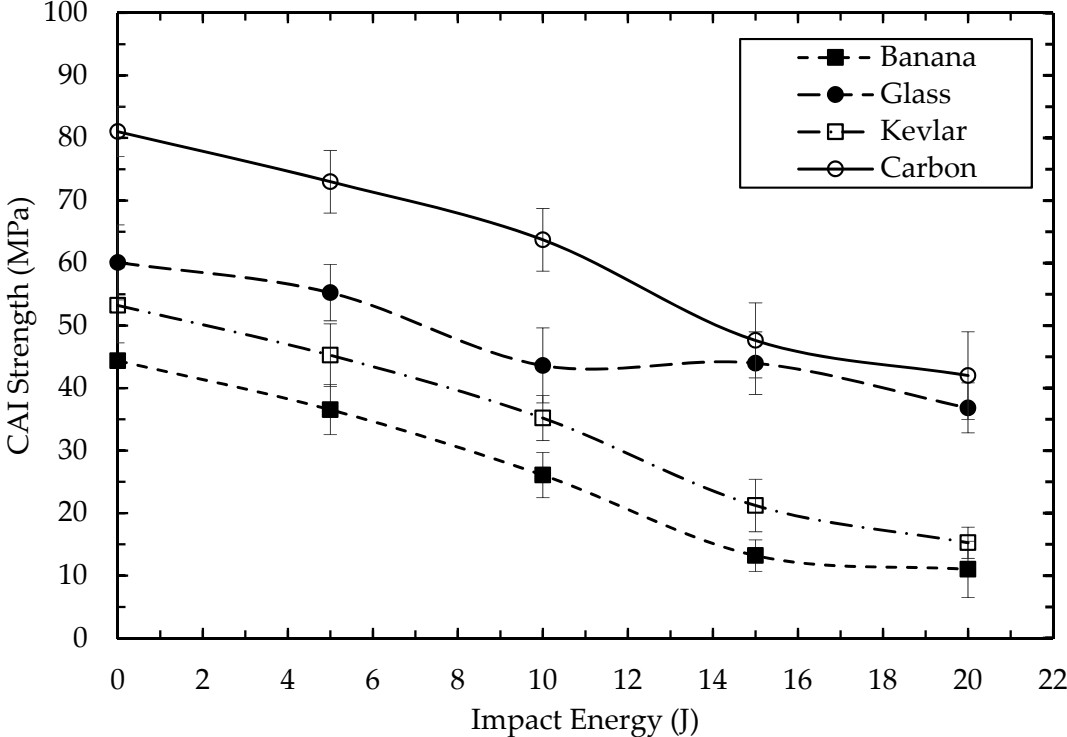

**Figure 12.** The average CAI strength versus impact energy traces of the sandwich structures.

In order to compare the properties of impact damage on the strength of the material, the ratio between the CAI strength of the specimens damaged by a given impact and the unimpacted sample was measured. Figure 13 presents the normalized residual strength as a function of the incident impact energy. The optimal banana/epoxy sandwich structure showed the greatest reduction in normalized strength; however, the lowest drop in the normalized residual strength was obtained in the carbon/epoxy core structures. In addition, the normalized CAI strength gradually decreased with the increasing incident impact energies. Similar findings were reported by Wang et al. [50]. At an impact energy of 15 J, the normalized reductions of this strength were: 41% in the carbon/epoxy structure, 43% in the glass/epoxy structure, 60% in the Kevlar/epoxy structure, and 70% in the optimum banana/epoxy structure. The lowest reduction in normalized value, especially of the synthetic fiber structure, may be due to the stiffer core, making the structure more stable and with a high resistance to impact. As a result, synthetic fiber/epoxy systems were more prone to buckling during CAI testing.

Figure 14 shows the CAI setup conditions of the optimum banana/epoxy sandwich structures subjected to a 15 J impact energy. The major delamination face was observed perpendicular to the loading direction. As can be seen in the figure, the failure form of the outer skin was more severe at the point of impact. In addition, the debonding phenomenon occurred between the core and the outer surface extended rapidly from the middle of the damaged area to both sides of the edge. All failure modes for the testing sample were similarly observed and were associated with the compression shear failure and local buckling at the point of impact [51].

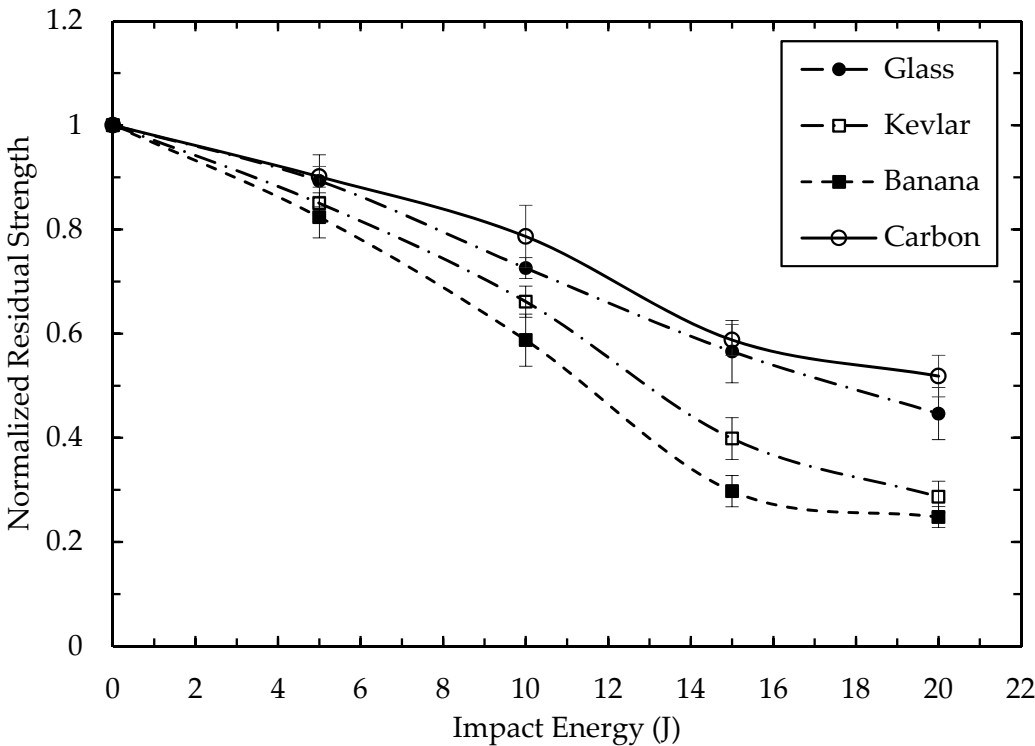

**Figure 13.** Normalized residual strength as a function of incident impact energy.

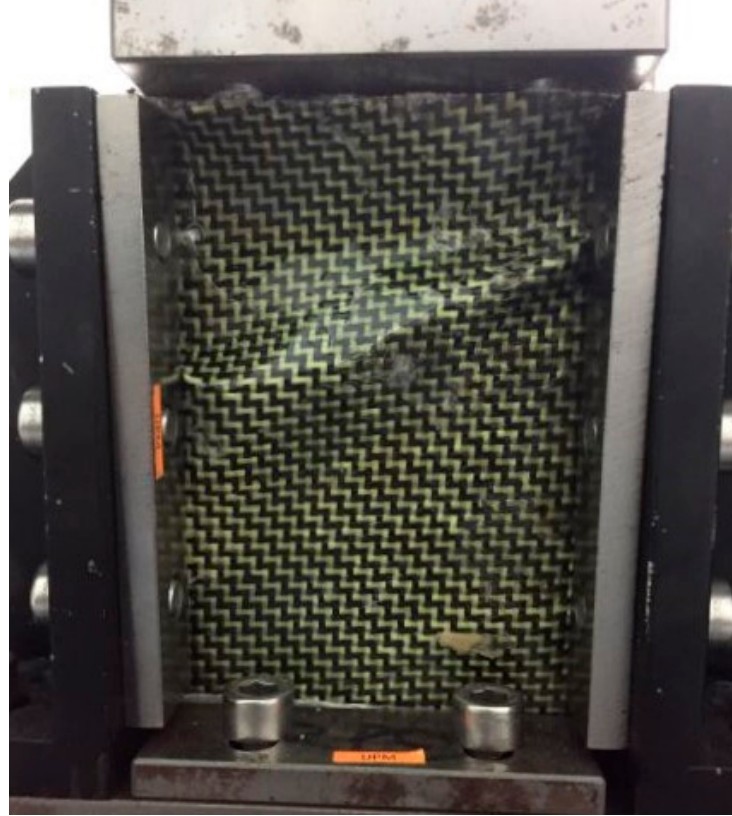

**Figure 14.** The CAI testing condition of the banana-epoxy sandwich structures subjected to an impact energy of 15 J.

Typical failure modes of the optimal banana, glass, carbon, and Kevlar epoxy sandwich structures subjected to compression loading are illustrated in Figure 15. It can be seen that the optimal banana epoxy core largely fractured from the middle position and the uppermost skin was folded (Figure 15a). From the side view of the failure specimen, compression shear cracks located at the impacted zone and delamination of the outer face sheet were extensively observed for the glass/epoxy system (Figure 15b). In addition, shear matrix cracking toward the direction of CAI loading was also obtained for the Kevlar structure, as shown in Figure 15c. This damage occurred due to stress concentration at the impact point of loading and the failure mode propagating throughout the area. The difference in the material properties between the high stiffness of the skin and the soft core contributed to the catastrophic failure of the sandwich structure during CAI testing [52]. Less core buckling and face sheet microbuckling of the carbon/epoxy structures was observed (Figure 15d), suggesting a high stiffness of the core and resistance to the axial compressive stress within the face sheets. V-shaped shear cracking was clearly apparent in this sandwich panel.

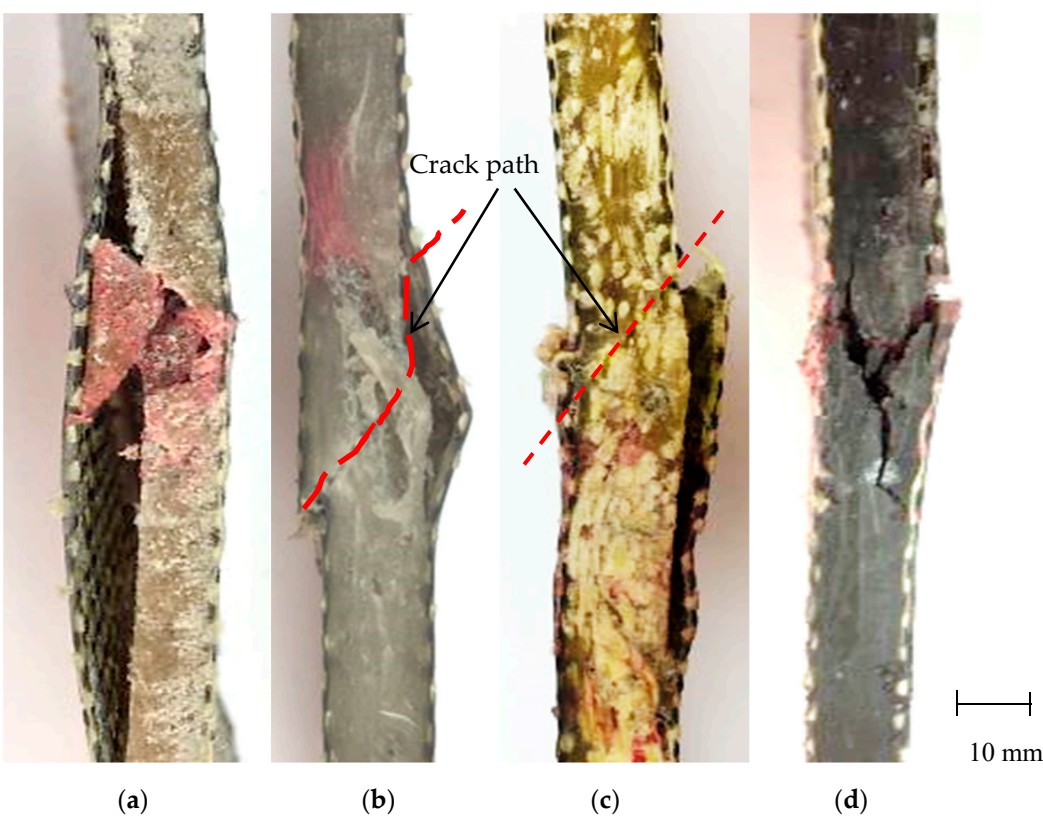

(**a**) (**b**) (**c**) (**d**)

**Figure 15.** Cross-section of failure modes for the CAI testing of the (**a**) banana, (**b**) glass, (**c**) carbon, and (**d**) Kevlar epoxy sandwich structures.

## 4. Conclusions

The low-velocity impact test and residual compression behaviors for the optimized fiber loading, fiber length, and alkaline treatment content of the banana-fiber reinforced sandwich structures were obtained. In this study, glass/epoxy, carbon/epoxy, and Kevlar/epoxy composites were also tested for comparison. This work contributes several key findings, as follows:

i.  The maximum stress and tensile modulus of the optimal banana-fiber-reinforced epoxy composite were increased up to 90% and 22%, respectively, compared to the epoxy-resin system.

ii. The effect of optimizing the compounding parameters of banana-reinforced epoxy was significantly comparable in terms of impact, damage tolerance, and residual impact due to higher bonding between fibers and matrices.

iii.    The low-velocity impact response of the optimal banana/epoxy composite gave a lower peak load. However, a significantly large damage area, higher energy absorption, and greater dent depth were obtained.

iv.    The banana fiber sandwich structure recorded a lower CAI resistance and a greater reduction in normalized strength than those using synthetic fibers.

v.    Delamination and core fracture were mainly observed in the optimal banana/epoxy structure. However, under the CAI testing matrix, shear cracking dominated in the Kevlar, carbon, and glass composites.

Finally, the results depicted significantly lower values for the impact resistance and residual damage tolerance of the optimum banana composite structures. Further investigation through matrix modification by adding nanofillers to natural fiber composites may yield interesting discoveries.

**Author Contributions:** Formal analysis, Z.A.R.; Funding acquisition, R.D. and M.Y.M.D.; Investigation, M.Z.H. and A.F.M.N.; Supervision, S.M.S. All authors have read and agreed to the published version of the manuscript.

**Funding:** This research was funded by Universiti Teknologi Malaysia under "Geran Universiti Penyelidik" (GUP) Tier 2 Scheme Q.K.130000.2656.15J85, Tier 2 Scheme Q.K.130000.2656.17J66, and UTMER Scheme Q.K.130000.2656.18J24.

**Acknowledgments:** The authors are thankful to the Ministry of Education Malaysia and Universiti Putra Malaysia for supporting this work through a Visiting Scholar (Post-Doctoral) scholarship.

**Conflicts of Interest:** The authors declare no conflict of interest.

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
