# Peer review of "Impact Damage Resistance and Post-Impact Tolerance of Optimum Banana-Pseudo-Stem-Fiber-Reinforced Epoxy Sandwich Structures"

_applsci, doi:10.3390/app10020684_

Round 1

Reviewer 1 Report

This work presents an interesting approach to the use of banana fiber as reinforcement of a polymer composite, compared with other synthetic fibers such as carbon, Kevlar or glass. Authors determine the mechanical behavior by low-velocity impact and compression after impact. The tests are performed under standards and the conclusions are according to the results. Only some remarks:

The explanation of the process for chopping the fiber to obtain the average length, that authors claim, should be extended. this is a key point to understand the efficiency of the composite. The same for the process for mixing the fiber with the epoxy resin.What's the level of uniformity in the distribution of the fiber in the matrix?any difference when using chopped carbon/glass/Kevlar fiber? Any cluster? Please justify the need of using a sandwich with external layers of textile of carbon/kevlar/glass Remove the lines 555 and 556 because these are coming from the template of the journal. In line 562 Bamboo?

Reviewer 2 Report

Pag 3 line 104 I suggest to the authors to add some refs or short explanation about the two methods (Taguchi and RSM) and to do the same things with the Box-Behnken method

Pag 5 line 216 I ask the authors to be clearer about the meaning of neat epoxy system, banana fiber composite, and optimum banana reinforced epoxy composites

I ask to the authors to add some indication about the standard deviation of the tensile results

Although in the abstract the authors anticipate that an optimization of the compounding parameters will be carried out, it is not very clear how and what parameters they have studied. I ask to the authors to be more explicit about that.
